# Learning from Rules Generalizing Labeled Exemplars

**Abhijeet Awasthi**  **Sabyasachi Ghosh**  **Rasna Goyal**  **Sunita Sarawagi**
Department of Computer Science and Engineering
Indian Instiute of Technology Bombay
Mumbai, Maharashtra 400076, India
`{awasthi,sghosh,goyalrasna,sunita}@cse.iitb.ac.in`

## Abstract

In many applications labeled data is not readily available, and needs to be collected via pain-staking human supervision. We propose a rule-exemplar method for collecting human supervision to combine the efficiency of rules with the quality of instance labels. The supervision is coupled such that it is both natural for humans and synergistic for learning. We propose a training algorithm that jointly denoises rules via latent coverage variables, and trains the model through a soft implication loss over the coverage and label variables. The denoised rules and trained model are used jointly for inference. Empirical evaluation on five different tasks shows that (1) our algorithm is more accurate than several existing methods of learning from a mix of clean and noisy supervision, and (2) the coupled rule-exemplar supervision is effective in denoising rules.

## 1 Introduction

With the ever-increasing reach of machine learning, a common hurdle to new adoptions is the lack of labeled data and the pain-staking process involved in collecting human supervision. Over the years, several strategies have evolved. On the one hand are methods like active learning and crowd-consensus learning that seek to reduce the cost of supervision in the form of per-instance labels. On the other hand is the rich history of rule-based methods (Appelt et al., 1993; Cunningham, 2002) where humans code-up their supervision as labeling rules. There is growing interest in learning from such efficient, albiet noisy, supervision (Ratner et al., 2016; Pal & Balasubramanian, 2018; Bach et al., 2019; Sun et al., 2018; Kang et al., 2018). However, clean task-specific instance labels continue to be critical for reliable results (Goh et al., 2018; Bach et al., 2019) in spite of easy availability of pre-trained models (Sun et al., 2017; Devlin et al., 2018).

In this paper we propose a unique blend of cheap coarse-grained supervision in the form of rules and expensive fine-grained supervision in the form of labeled instances. Instead of supervising rules and instance labels independently, we propose that each labeling rule be attached with exemplars of where the rule correctly 'fires'. Thus, the rule can be treated as a noisy generalization of those exemplars. Often rules are coded up only after inspecting data. As a human inspects instances, he labels them, and then generalizes them to rules. Thus, humans provide paired supervision of rules and exemplars demonstrating correct deployment of that rule. We explain further with two illustrative applications. Our examples below are from the text domain because rules have been traditionally used in many NLP tasks, but our learning algorithm is agnostic to how rules are expressed.

**Sentiment Classification** Consider an instance `I highly recommend this modest priced cellular phone` that a human inspects for a sentiment labeling task. After labeling it as `positive`, he can easily generalize it to a rule `Contains 'highly recommend' →` `positive label`. This rule generalizes to several more instances, thereby eliminating the need of per-instance labeling on those. However, the label assigned by this rule on unseen instances may not be as reliable as the explicit label on this specific exemplar it generalized. For example, it misfires on `I would highly recommend this phone if it weren't for their poor service.`

---

Code and datasets available at https://github.com/awasthiabhijeet/Learning-From-Rules

**Slot-filling** Consider a slot-filling task on restaurant reviews over labels like `cuisine,` `location,` and `time.` When an annotator sees an instance like: `what chinese` `restaurants in this city have good reviews?`, after labeling token `chinese` as `cuisine,` he generalizes it to a rule: `(.*ese|.*ian|mexican) restaurants →` `(cuisine) restaurants.` This rule matches hundreds of instances in the unlabeled set, but could wrongly label a phrase like `these restaurants.` Our focus in this paper is developing algorithms for training models under such coupled rule-exemplar supervision. Our main challenge is that the labels induced by the rules are more noisy than instance-level supervised labels because humans tend to over generalize (Tessler & Goodman, 2019) as we saw in the illustrations above. Learning with noisy labels with or without additional clean data has been a problem of long-standing interest in ML (Khetan et al., 2018; Zhang & Sabuncu, 2018; Ren et al., 2018b; Veit et al., 2017; Shen & Sanghavi, 2019). However, we seek to design algorithms that better capture rule-specific noise with the help of exemplars around which we have supervision that the rule fired correctly. We associate a latent random variable on whether a rule correctly 'covers' an instance, and jointly learn the distribution among the label and all cover variables. This way we simultaneously train the classifier with corrected rule-label examples, and restrict over-generalized rules. The denoised rules are used during inference to further boost accuracy of the trained model. In summary our contributions in this paper are as follows:

**Our contributions** (1) We propose the paradigm of supervision in the form of rules generalizing labeled exemplars that is natural in several applications. (2) We design a training method that simultaneously denoises over-generalized rules via latent coverage variables, and trains a classification model with a soft implication loss that we introduce. (3) Through experiments on five tasks spanning question classification, spam detection, sequence labeling, and record classification we show that our proposed paradigm of supervision enables an effective synergy between rule-level and instance-level supervision. (4) We compare our algorithm to several recent frameworks for learning with noisy supervision and constraints, and show much better results with our method.

## 2    TRAINING WITH RULES AND EXEMPLARS

We first formally describe the problem of learning from rules generalizing examplars on a classification task. Let $\mathcal{X}$ denote the space of instances and $\mathcal{Y} = \{1, \ldots, K\}$ denote the space of class labels. Let the set of labeled examples be $L = \{(\mathbf{x}_1, \ell_1, e_1), \ldots, (\mathbf{x}_n, \ell_n, e_n)\}$ where $\mathbf{x}_i \in \mathcal{X}$ is an instance, $\ell_i \in \mathcal{Y}$ is its user-provided label, and $e_i \in \{R_1, \ldots, R_m, \emptyset\}$ denotes that $\mathbf{x}_i$ is an exemplar for rule $e_i$. Some labeled instances may not be generalized to rules and for them $e_i = \emptyset$. Also, a rule can have more than one exemplar associated with it. Each rule $R_j$ could be a blackbox function $R_j : \mathbf{x} \mapsto \{\ell_j, \varnothing\}$ that takes as input an instance $\mathbf{x} \in \mathcal{X}$ and assigns it either label $\ell_j$ or no-label. When the $i$th labeled instance is an exemplar for rule $R_j$ (that is, $e_i = R_j$), the label of the instance $\ell_i$ should be $\ell_j$. Additionally, we have a different set of unlabeled instances $U = \{\mathbf{x}_{n+1}, \ldots, \mathbf{x}_N\}$. The cover set $H_j$ of rule $R_j$ is the set of all instances in $U \cup L$ for which $R_j$ assigns a noisy label $\ell_j$. An instance may be covered by more than one rule or no rule at all, and the labels provided by these rules may be conflicting. Our goal is to train a classification model $P_\theta(y|\mathbf{x})$ using $L$ and $U$ to maximize accuracy on unseen test instances. A baseline solution is to use $R_j$ to noisily label the covered $U$ instances using majority or other consensus method of resolving conflicts. We then train $P_\theta(y|\mathbf{x})$ on the noisy labels using existing algorithms for learning from noisy and clean labels (Veit et al., 2017; Ren et al., 2018b). However, we expect to be able to do better by learning the systematic pattern of noise in rules along with the classifier $P_\theta(y|\mathbf{x})$.

**Our noise model on $R_j$** A basic premise of our learning paradigm is that the noise induced by a rule $R_j$ is due to *over-generalizing* the exemplar(s) seen when creating the rule. And, there exists a smaller neighborhood closer to the exemplar(s) where the noise is zero. We model this phenomenon by associating a latent Bernoulli random variable $r_{ji}$ for each instance $\mathbf{x}_i$ in the stated cover set $H_j$ of each rule $R_j$. When $r_{ji} = 1$, rule $R_j$ has not over-generalized on $\mathbf{x}_i$, and there is no noise in the label $\ell_j$ that $R_j$ assigns to $\mathbf{x}_i$. When $r_{ji} = 0$ we flag an over-generalization, and abstain from labeling $\mathbf{x}_i$ as $\ell_j$ suspecting it to be too noisy. We call $r_{ji}$s as the latent coverage variables. We propose to learn the distribution of $r_j$ using another network with parameters $\phi$ that outputs the probability $P_{j\phi}(r_j|\mathbf{x})$ that $r_j = 1$. We then seek to jointly learn $P_\theta(y|\mathbf{x})$ and $P_{j\phi}(r_j|\mathbf{x})$ to model the distribution over the true label $y$ and true coverage $r_j$ for each rule $j$ and each $\mathbf{x}$ in $H_j$. Thus

$P_{j\phi}$ plays the role of restricting a rule $R_j$ so that $r_j$ is not necessarily 1 for all instances in its cover set $H_j$

**An example** We make our discussion concrete with an example. Figure 1 shows a two-dimensional $\mathcal{X}$ space with labeled points $L$ denoted as red crosses and blue circles, unlabeled points as dots, and the true labels as background color of the region. We show two rule-exemplar pairs: $(\mathbf{x}_1, y_1 = \text{red}, R_1), (\mathbf{x}_2, y_2 = \text{blue}, R_2)$ with bold boundaries. Clearly, both rules $R_1, R_2$ have over-generalized to the wrong region. If we train a classifier with many examples in $H_1 \cup H_2$ wrongly labeled by rules, then even with a noise tolerant loss function like Zhang &

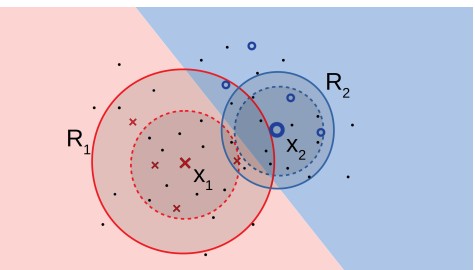

Figure 1: Restricting over-generalized rules

Sabuncu (2018), the classifier $P_\theta(y|\mathbf{x})$ might be misled. In contrast, what we hope to achieve is to learn the $P_{j\phi}(r_j|\mathbf{x})$ distribution using the limited labeled data and the overlap among the rules such that $\Pr(r_j|\mathbf{x})$ predicts a value of 0 for examples wrongly covered. Such examples are then excluded from training $P_\theta$. The dashed boundaries indicate the revised boundaries of $R_j$s that we can hope to learn based on consensus on the labeled data and the set of rules. Even after such restriction, $R_j$s are useful for training the classifier because of the unlabeled points inside the dashed regions that get added to the labeled set.

## 2.1 How we Jointly learn $P_\theta$ and $P_{j\phi}$

In general we will be provided with several rules with arbitrary overlap in the set of labeled $L$ and unlabeled examples $U$ that they cover. Intuitively, we want the label distribution $P_\theta(y|\mathbf{x})$ to correctly restrict the coverage distribution $P_{j\phi}(r_j|\mathbf{x})$, which in turn can provide clean labels to instances in $U$ that can be used to train $P_\theta(y|\mathbf{x})$. We have two types of supervision in our setting. First, individually for each of $P_\theta(y|\mathbf{x})$ and $P_{j\phi}(r_j|\mathbf{x})$ we have ground truth values of $y$ and $r_j$ for some instances. For the $P_\theta(y|\mathbf{x})$ distribution, supervision on $y$ is provided by the human labeled data $L$, and we use these to define the usual log-likelihood as one term in our training objective:

$$\max_\theta LL(\theta) = \max_\theta \sum_{(\mathbf{x}_i, \ell_i) \in L} \log P_\theta(\ell_i | \mathbf{x}_i) \tag{1}$$

For learning the distribution $P_{j\phi}(r_j|\mathbf{x})$ over the coverage variables, the only sure-shot labeled data is that $r_{ji} = 1$ for any $\mathbf{x}_i$ that is an exemplar of rule $R_j$ and $r_{ji} = 0$ for any $\mathbf{x}_i \in H_j$ whose label $\ell_i$ is different from $\ell_j$. For other labeled instances $\mathbf{x}_i$ covered with rules $R_j$ with agreeing labels, that is $\ell_i = \ell_j$ we do not strictly require that $r_{ji} = 1$. In the example above the corrected dashed red boundary excludes a red labeled point to reduce its noise on other points. However, if the number of labeled exemplars are too few, we regularize the networks towards more rule firings, by adding a noise tolerant $r_{ji} = 1$ loss on the instances with agreeing labels. We use the generalized cross entropy loss of Zhang & Sabuncu (2018).

$$LL(\phi) = \sum_{(\mathbf{x}_i, \ell_i, e_i) \in L} \Big( \log P_{e_i \phi}(r_{e_i i} = 1 | \mathbf{x}_i) + \sum_{j: \mathbf{x}_i \in H_j \wedge \ell_i \neq \ell_j} \log P_{j\phi}(r_{ji} = 0 | \mathbf{x}_i)$$
$$- \sum_{j: \mathbf{x}_i \in H_j \wedge \ell_i = \ell_j} \text{Generalized-XENT}(P_{j\phi}(r_j | \mathbf{x}_i), r_{ji} = 1) \Big) \tag{2}$$

Note for other instances $\mathbf{x}_i$ in $R_j$'s cover $H_j$, value of $r_{ji}$ is unknown and latent. The second type of supervision is on the relationship between $r_{ji}$ and $y_i$ for each $\mathbf{x}_i \in H_j$. A rule $R_j$ imposes a causal constraint that when $r_{ji} = 1$, the label $y_i$ has to be $\ell_j$.

$$r_{ji} = 1 \implies y_i = \ell_j \quad \forall \mathbf{x}_i \in H_j \tag{3}$$

We convert this hard constraint into a (log) probability of the constraint being satisfied under the $P_\theta(y|\mathbf{x})$ and $P_{j\phi}(r_j|\mathbf{x})$ distributions as:

$$\log \big( 1 - P_{j\phi}(r_j = 1 | \mathbf{x})(1 - P_\theta(\ell_j | \mathbf{x})) \big) \tag{4}$$

Figure 2 shows a surface plot of the above log probability as a function of $P_\theta(\ell_j | \mathbf{x})$ (shown as axis P(y) in figure) and $P_{j\phi}(r_j = 1 | \mathbf{x})$ (shown as axis P(r) in figure) for a single rule.

Observe that likelihood drops sharply as $P(r_j|\mathbf{x})$ is close to 1 but $P(y = \ell_j|\mathbf{x})$ is close to zero. For all other values of these probabilities the log-likelihood is flat and close to zero. Specifically, when $P_{j\phi}$ predicts low values of $r_j$ for a $\mathbf{x}$, the log-likelihood surface is flat, effectively withdrawing the $(\mathbf{x}, \ell_j)$ supervision from training the classifier $P_\theta$. Thus maximizing this likelihood provides a soft enforcement of the constraint without unwanted biases. We call this the negative *implication loss*.

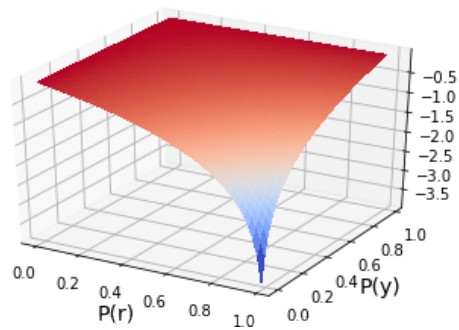

Figure 2: Negative implication loss

We do not need to explicitly model the conflict among rules, that is when an $\mathbf{x}_i$ is covered by two rules $R_j$ and $R_k$ of differing labels ($\ell_j \neq \ell_k$), then both $r_{ji}$ and $r_{ki}$ cannot be 1. This is because the constraint among pairs $(y_i, r_{ji})$ and $(y_i, r_{ki})$ as stated in Equation 3 subsumes this one.

During training we then seek to maximize the log of the above probability along with normal data likelihood terms. Putting the terms in Equations 1, 2 and 4 together our final training objective is:

$$\min_{\theta,\phi} -LL(\theta) - LL(\phi) - \gamma \sum_{j;\mathbf{x} \in H_j \cap U} \log(1 - P_{j\phi}(r_j = 1|\mathbf{x})(1 - P_\theta(\ell_j|\mathbf{x}))) \tag{5}$$

We refer to our training loss as a denoised rule-label implication loss or ImplyLoss for short. The $LL(\phi)$ term seeks to denoise rule coverage which then influence the $y$ distribution via the implication loss. We explored several other methods of enforcing the constraint among $y$ and $r_j$ in the training of the $P_\theta$ and $P_{j\phi}$ networks. Our method ImplyLoss consistently performed the best among several methods we tried including the recent posterior regularization (Ganchev et al., 2010; Hu et al., 2016) method of enforcing soft constraints and co-training (Blum & Mitchell, 1998).

**Network Architecture** Our network has three modules. (1) A shared embedding layer that provides the feature representation of the input. When labeled data is scarce, this will typically be a pre-trained layer from a related task. The embedding module is task-specific and is described in the experiment section. (2) A classification network that models $P_\theta(y|\mathbf{x})$ with parameters $\theta$. The embedding of an input $\mathbf{x}$ is passed through multiple non-linear layers with ReLU activation, a last linear layer followed by Softmax to output a distribution over the class labels. (3) A rule network that models $P_{j\phi}(r_j = 1|\mathbf{x})$ whose parameters $\phi$ are shared across all rules. The input to the network is rule-specific and concatenates the embedding of the input instance $\mathbf{x}$, and a one-hot encoding of the rule id 'j'. The input is passed through multiple non-linear layers with ReLU activation before passing through a Sigmoid activation which outputs the probability $P_{j\phi}(r_j = 1|\mathbf{x})$.

**Inference** During prediction, joint inference over the label $y$ and coverage variables $r_j$ provides slight gains over depending solely on $P_\theta(y|\mathbf{x})$. For any test example $\mathbf{x}$, consider the set of rules $G$ covering $\mathbf{x}$ such that $P_{j\phi}(1|\mathbf{x}) > 0.5$. Probabilities from the label and coverage variables are combined to obtain a score $s(y)$ for each label $y$ as:

$$s(y|\mathbf{x}) = P_\theta(y|\mathbf{x}) + \frac{\sum_{R_j \in G} \delta(\ell_j = y)P_{j\phi}(1|\mathbf{x}) + \delta(\ell_j \neq y)P_{j\phi}(0|\mathbf{x})}{|G|} \tag{6}$$

The above can be viewed as a soft voting over the trained classifier $P_\theta$ and labels provided by rules with uncertain coverage. Because we also learned to denoise rules along with training the classifier, the labels assigned by the rules have higher precision than original rules.

## 3  EXPERIMENTS

We compare our training algorithms against simple baselines, existing error-tolerant learning algorithms, and existing constraint-based learning in deep networks.

We evaluate across five datasets spanning three task types: text classification, sequence labeling, and record classification. We augment the datasets with rules, that we obtained manually in three

| Dataset | $|L|$ | $|U|$ | #Rules | %Cover | Precision | %Conflict | Avg $|H_j|$ | #Rules Per Instance | |Valid| | |Test| |
|---|---|---|---|---|---|---|---|---|---|---|
| Question | 68 | 4884 | 68 | 95 | 63.8 | 22.5 | 124 | 1.8 | 500 | 500 |
| MIT-R | 1842 | 64888 | 15 | 14 | 80.7 | 2.5 | 634 | 1.1 | 4091 | 14256 |
| SMS | 69 | 4502 | 73 | 40 | 97.3 | 0.6 | 31 | 1.3 | 500 | 500 |
| YouTube | 100 | 1586 | 10 | 87 | 78.6 | 30.2 | 258 | 1.9 | 120 | 250 |
| Census | 83 | 10000 | 83 | 100 | 84.1 | 27.5 | 540 | 4.5 | 5561 | 16281 |

Table 1: Statistics of datasets and their rules. %Cover is fraction of instances in $U$ covered by at least one rule. Precision refers to micro precision of rules. Conflict denotes the fraction of instances covered by conflicting rules among all the covered instances. Avg $|H_j|$ is average cover size of a rule in $U$. Rules Per Instance is average number of rules covering an instance in $U$.

cases, from pre-existing public sources in one case, and automatically in another. Table 1 presents statistics summarizing the datasets and rules. A brief description of each appears below.

**Question Classification** (Li & Roth, 2002): This is a TREC-6 dataset to classify a question to one of six categories: {`Abbreviation, Entity, Description, Human, Location, Numeric-value`}. The training set has 5452 instances which are split as 68 for $L$, 500 for validation, and the remaining as $U$. Each example in $L$ is generalized as a rule represented by a regular expression. E.g. After labeling `How do you throw a housewarming party ?` as `Description` we define a rule

$$(\text{how}|\text{How}|\text{what}|\text{What})(\text{does}|\text{do}|\text{to}|\text{can}).* \rightarrow \text{Description}.$$

More rules in Table 4 of supplementary. Although, creating such 68 generalised rules required 90 minutes, the generalizations cover 4637 instances in $U$, almost two orders of magnitude more instances than in $L$! On an average each of our rule covered 124 instances ($|H_j|$ column in Table 1). But the precision of labels assigned by rules was only 63.8%. 22.5% of covered instances had an inter-rule conflict, demonstrating noise in the rule labelings. Accuracy is used as the performance metric.

**MIT-R**[1] (Liu et al., 2013): This is a slot-filling task on sentences about restaurant search and the task is to label each token as one of {`Location, Hours, Amenity, Price, Cuisine, Dish, Restaurant_Name, Rating, Other`}. The training data is randomly split into 200 sentences (1842 tokens) as $L$, 500 sentences (4k tokens) as validation and remaining 6.9k sentences (64.9k tokens) as $U$. We manually generalize 15 examples in $L$. E.g. After inspecting the sentence `where can i get the highest rated burger within ten miles` and labeling `highest rated` as `Rating`, we provide the rule:

$$.*(\text{highly}|\text{high}|\text{good}|\text{top}|\text{highest})(\text{rate}|\text{rating}|\text{rated}).* \rightarrow \text{Rating}$$

to the matched positions. More examples in Table 7 of supplementary. Although, creating 15 generalizing rules took 45 minutes of annotator effort, the rules covered roughly 9k tokens in $U$. F1 metric is used for evaluation on the default test set of 14.2k tokens over 1.5k sentences.

**SMS Spam Classification** (Almeida et al., 2011): This dataset contains 5.5k text messages labeled as spam/not-spam, out of which 500 were held out for validation and 500 for testing. We manually generalized 69 exemplars to rules. Remaining examples go in the $U$ set. The rules here check for presence of keywords or phrases in the SMS `.* guaranteed gift .* → spam`. A rule covers 31 examples on an average and has a precision of 97.3%. However, in this case only 40% of the unlabeled set is covered by a rule. We report F1 here since class is skewed. More examples in Table 5 of supplementary.

**Youtube Spam Classification** (Alberto et al., 2015): Here the task is to classify comments on YouTube videos as Spam or Not-Spam. We obtain this from Snorkel's Github page[2], which provides 10 labeling functions which we use as rules, an unlabeled train set which we use as $U$, a labeled dev set to guide the creation of their labeling functions which we use as $L$, and labeled test and validation sets which we use in the same roles. Their labeling functions have a large coverage (258 on average), and a precision of 78.6%.

**Census Income** (Dua & Graff, 2019): This UCI dataset is extracted from the 1994 U.S. census. It lists a total of 13 features of an individual such as age, education level, marital status, country of

---

[1] groups.csail.mit.edu/sls/downloads/restaurant/
[2] https://github.com/snorkel-team/snorkel-tutorials/tree/master/spam

| Methods | Datasets | | | | |
|---|---|---|---|---|---|
| | Question (Accuracy) | MIT-R (F1) | YouTube (Accuracy) | SMS (F1) | Census (Accuracy) |
| Majority (No parameters trained) | 60.9 (0.7) | 40.9 (0.1) | 82.2 (0.9) | 48.4 (1.2) | 80.1 (0.1) |
| Only-L | 72.9 (0.6) | 73.5 (0.3) | 90.9 (1.8) | 89.0 (1.6) | 79.4 (0.5) |
| L+Umaj | - 1.4 (1.5) | + 0.0 (0.3) | + 0.8 (1.9) | + 3.5 (1.2) | + 0.9 (0.1) |
| Noise-tolerant (Zhang et al., 2018) | - 0.5 (1.1) | + 0.0 (0.2) | + 1.7 (1.1) | + 2.9 (1.2) | + 1.0 (0.2) |
| L2R (Ren et al., 2018b) | + 0.3 (2.1) | - 15.4 (1.0) | + 2.5 (0.5) | + 2.3 (0.8) | **+ 2.9** (0.3) |
| L+Usnorkel (Ratner et al., 2016) | - 0.7 (3.0) | + 0.0 (0.2) | + 2.7 (0.7) | + 3.5 (1.3) | + 1.0 (0.4) |
| Snorkel-Noise-Tolerant | - 1.4 (1.6) | + 0.0 (0.3) | + 2.0 (0.7) | + 2.7 (1.5) | + 0.2 (0.5) |
| Posterior Reg. (Hu et al., 2016) | - 0.8 (1.0) | - 0.1 (0.4) | - 2.9 (1.9) | + 1.8 (1.5) | - 0.8 (0.5) |
| ImplyLoss (**Ours**) | **+ 11.7** (1.5) | **+ 0.8** (0.3) | **+ 3.2** (1.1) | **+ 4.2** (1.0) | + 1.7 (0.2) |

Table 2: Comparison of ImplyLoss (our method) with various methods (described in Section 3.1) on five different datasets. The numbers reported for all methods after the double-line are gains over the baseline (Only-L) that does not use rules at all. Higher is better. **NOTE**: Numbers in brackets represent standard deviation of the original accuracy and not of gains.

origin etc. The primary task on it is binary classification - whether a person earns more than \$50K or not. The train data consists of 32563 records. We choose 83 random data points as $L$, 10k points as $U$ and 5561 points as validation data. For this case we created the rules synthetically as follows: We hold out disjoint 16k random points from the training dataset as a proxy for human knowledge and extract a PART decision list (Frank & Witten, 1998) from it as our set of rules. We retain only those rules which fire on $L$.

**Network Architecture**  Since our labeled data is small we depend on pre-trained resources. As the embedding layer we use a pretrained ELMO (Peters et al., 2018) network where 1024 dimensional contextual token embeddings serve as representations of tokens in the MIT-R sentences, and their average serve as representation for sentences in Question and SMS dataset. Parameters of the embedding network are held fixed during training. For sentences in the YouTube dataset, we use Snorkel's[2] architecture of a simple bag-of-words feature representation marking the frequent unigrams and bi-grams present in a sentence using a few-hot vector. For the Census dataset categorical features are represented as one hot vectors, while real valued features are simply normalized. For MIT-R, Question and SMS both classification and rule-weight network contain two 512 dimensional hidden layers with ReLU activation. For Census, both the networks contain two 256 dimensional hidden layers with ReLU activation. For YouTube, the classifier network is a simple logistic regression like in Snorkel's code. The rule network has one 32-dimensional hidden layer with ReLU activation.

Each reported number is obtained by averaging over ten random initializations. Whenever a method involved hyper-parameters to weigh the relative contribution of various terms in the objective, we used a validation dataset to tune the value of the hyper-parameter. Hyperparameters used are provided in Section C of supplementary.

## 3.1 Comparison with different methods

In Table 2 we compare our method with the following alternatives on each of the five datasets:

**Majority:** that predicts via majority vote among the rules that cover an instance. This baseline indicates the stand-alone quality of rules, no network is learned here. Ties are broken arbitrarily for class-balanced datasets or by using a default class. Table 2, shows that the accuracy of majority is quite poor indicating either poor precision or poor coverage of the rule sets.[3].

**Only-L**: Here we train the classifier $P_\theta(y|\mathbf{x})$ only on the labeled data $L$ using the standard cross-entropy loss (Equation 1). Rule generalisations are not utilized at all in this case. We observe in Table 2 that even with the really small labeled set we used for each dataset, the accuracy of a classifier learned with clean labeled data is much higher than noisy majority labels of rules. We consider this method as our baseline and report the gains on remaining methods.

---

[3]Only for the Census dataset the relative accuracy is high because the rules were obtained synthetically through a rule-learning algorithm on a very large labeled dataset to serve as a proxy for a human's generalization.

**L+Umaj**: Next we train the classifier on $L$ along with $U_{\text{maj}}$ obtained by labeling instances in $U$ with the majority label among the rules applicable to the instance. Loss corresponding to the examples labeled by rules is weighted as follows:

$$\min_{\theta} \sum_{(\mathbf{x}_j, \ell_j) \in L} -\log P_{\theta}(\ell_j | \mathbf{x}_j) + \gamma \sum_{(\mathbf{x}_j, y_j) \in U_{\text{maj}}} -\log P_{\theta}(y_j | \mathbf{x}_j) \qquad (7)$$

The row corresponding to L+Umaj in Table 2 provides the gains of this method over Only-L. We observe gains with the noisily labeled $U$ in three out of the five cases.

**Noise-tolerant:** Since labels in $U_{\text{maj}}$ are noisy, we next use Zhang & Sabuncu (2018)'s noise tolerant generalized cross entropy loss on them with regular cross-entropy loss on the clean $L$ as follows:

$$\min_{\theta} \sum_{(\mathbf{x}_j, \ell_j) \in L} -\log P_{\theta}(\ell_j | \mathbf{x}_j) + \gamma \sum_{(\mathbf{x}_j, y_j) \in U_{\text{maj}}} \frac{(1 - P_{\theta}(y_j | \mathbf{x}))^q}{q} \qquad (8)$$

Parameter $q \in [0, 1]$ controls the noise tolerance which we tune as a hyper-parameter. We observe that in three cases minimizing the above objective improves beyond L+Umaj validating that noise-tolerant loss functions can be useful for learning from noisy labels on $U_{\text{maj}}$.

**Learning to Reweight (L2R)** (Ren et al., 2018b): is a recent method for training with a mix of clean and noisy labeled data. They train the classifier by meta-learning to re-weight the loss on the noisily labelled instances ($U_{\text{maj}}$) with the help of the clean examples ($L$). This method provides significant accuracy gains over Only-L in three out the five datasets. However, it fails in the multi-class classification task of slot-filling which has a very high class imbalance and rules of smaller coverage.

All the above methods employ no extra parameters to denoise or weight individual rules. We next compare with a number of methods that do.

**L+Usnorkel:** This method replaces Majority-based consensus with Snorkel's generative model (Ratner et al., 2016) that assigns weights to rules and labels examples in $U$. Thereafter we use the same approach as in L+Umaj with just Snorkel's soft-labels instead of Majority on $U$. We also compare with using noise-tolerant loss on $U$ labeled by Snorkel (Eqn:8) which we call **Snorkel-Noise-Tolerant**. Like previous methods, both of these methods provide improvements over Only-L on three of the five datasets where the rules are less noisy. L+Usnorkel performs slightly better than Noise-Tolerant on $U_{\text{maj}}$.

We next compare with a method that simultaneously learns two sets of networks $P_{\theta}$ and $P_{j\phi}$ like ours but with different loss function and training schedule.

**Posterior Regularization (PR):** This method proposed in Hu et al. (2016) also treats rules as soft-constraints and has been used for training neural networks for structured outputs. They use Ganchev et al. (2010)'s posterior regularization framework to train the two networks in a teacher-student setup. We adapt the same framework and get a procedure as follows: The student proposes a distribution over $y$ and $r_j$s using current $P_{\theta}$ and $P_{j\phi}$, the teacher uses the constraint in Eq 3 to revise the distributions so as to minimize the probability of violations, the student updates parameters $\theta$ and $\phi$ to minimize KL distance with the revised distribution. The detailed formulation appear in the Section A of supplementary. We find that this method is no better than Only-L in most of the cases and worse than the noise-tolerant method that does not train extra $\phi$ parameters.

**ImplyLoss(Ours)**: Overall our approach of training with denoised rule-label implication loss provides much better accuracy than all the above eight methods and we get consistent gains over Only-L on all datasets. On the Question dataset we get 11.7 points gain over Only-L whereas the best gain by existing method was 0.3. A useful property of our method compared to the PR method above is that the training process is simple and fits into the batch stochastic gradient training template. In contrast, PR requires special alternating computations. We next perform a number of diagnostics experiments to explain the reasons for the superior performance of our method.

**Diagnostics: Effectiveness of learning true coverage via $P_{j\phi}$** An important part of our method is the rule-specific denoising learned via the $P_{j\phi}$ network. In the chart alongside we plot the original precision of rules on the test data, and the precision after suppressing those rule labelings where $P_{j\phi}(r_j|\mathbf{x})$ predicts 0 instead of 1. Observe now that the precision is more than 91% on all datasets. For the Question dataset, the precision jumped from 64% to 98%. The percentage of labelings suppressed (shown by the dashed line) is higher on datasets with noisier rules (e.g. compare Question and

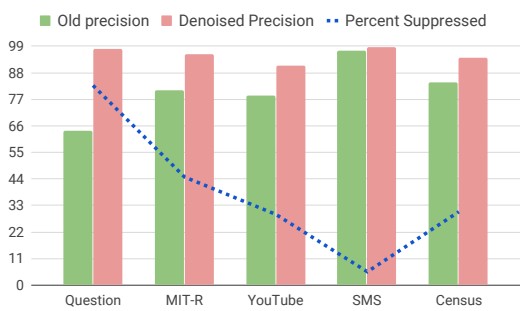

Figure 3: Rule-specific denoising by our method.

SMS). This shows that $P_{j\phi}$ is able to denoise rules by capturing the distribution of the latent true coverage variables with the limited $LL(\phi)$ loss and indirectly via the implication loss.

**Effect of rule precision** Rules in the Census dataset are of higher quality in terms of precision as well as coverage. Superior performance of the L2R method on this dataset motivated us to inspect how well our method performs on the same dataset in the absence of high precision rules. We created four new versions of the rule sets by successively removing high precision rules from the original rule set. We observe that our method performs better than L2R when rules have low precision. Because Imply-Loss denoises rules, it is better able to handle low-precision rules.

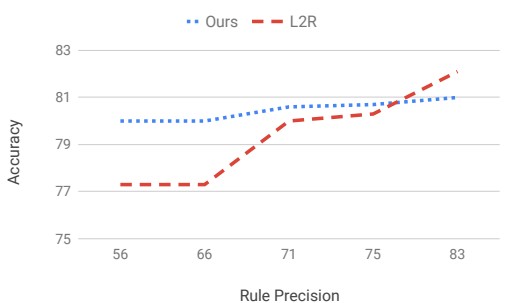

Figure 4: Effect of rule precision

**Role of Exemplars in Rules** We next evaluate the importance of the exemplar-rule pairs in learning the $P_{j\phi}$ and $P_\theta$ networks. The exemplars of a rule give an interesting new form of supervision about an instance where a labeling rule must fire. To evaluate the importance of this supervision, we exclude the $r_j = 1$ likelihood on rule-exemplar pairs from $LL(\phi)$, that is, the first term in Equation 2 is dropped. In the table below we see that performance of ImplyLoss usually drops when the exemplar-rule supervision is removed. Interestingly, even after this drop, the performance of ImplyLoss surpasses most of the methods in Table 2 indicating that even without exemplar-rule pairs our training objective is effective in learning from rules and labeled instances.

| | Question | MIT-R | SMS | Census |
|---|---|---|---|---|
| $r_j = 1$ for rule-exemplar pairs | 84.5 (1.5) | 73.7 (0.3) | 93.2 (1.0) | 81.0 (0.2) |
| No $r_j = 1$ for rule-exemplar pairs | 83.8 (0.7) | 73.5 (0.5) | 93.5 (1.2) | 80.8 (0.3) |

Table 3: Effect of removing rule-exemplar supervision from $LL(\phi)$

**Effect of increasing labeled data $L$** We increase $L$ while keeping the number of rules fixed on the Question dataset. In the attached plot we see the accuracy of our method (Imply-Loss) against Only-L, L+Usnorkel and Posterior Reg. We observe the expected trend that the gap between the method narrows as labeled data increases.

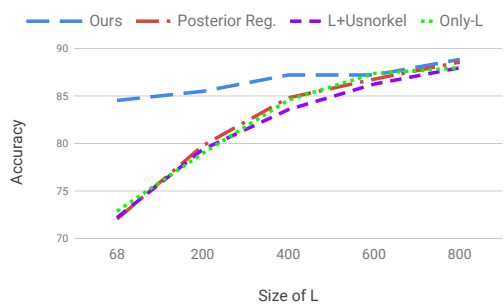

Figure 5: Effect of increasing labeled data

## 4 RELATED WORK

Learning from noisily labeled data has been extensively studied in settings like crowdsourcing. One category of these algorithms upper-bound the loss function to make it robust to noise. These include methods like MAE (Ghosh

et al., 2017), Generalized Cross Entropy (CE)(Zhang & Sabuncu, 2018), and Ramp loss (Collobert et al., 2006). Most of these assume that noise is independent of the input given the true label. In our model noise is systematic and instance-dependent.

A second category assume that a small clean dataset is available along with noisily labeled data. This is also true in our case, and we compared with a state of the art method in that category Ren et al. (2018b) that chooses a descent direction that aligns with a clean validation set using meta-learning. Others in this category include: Shen & Sanghavi (2019)'s method of iteratively selecting examples with smallest loss, and Veit et al. (2017)'s method of learning a separate network to transform noisy labels to cleaned ones which are used to impose a cross-entropy loss on $P_\theta(y|\mathbf{x})$. In contrast, we perform rule-specific cleaning via latent coverage variables and a flexible implication loss which withdraws $y$ supervision when $P_{j\phi}(r_{ji}|\mathbf{x})$ assumes low values. Another way of relating clean and noisy labels is via an instance-independent confusion matrix learned jointly with the classifier (Khetan et al., 2018; Goldberger & Ben-Reuven, 2016; Han et al., 2018b;a). These works assume that the confusion matrix is instance independent, which does not hold for our case. Tanaka et al. (2018) uses confidence from the classifier to eliminate noise but they need to ensure that the network does not memorize noise. Our learning setup also has the advantage of extracting confidence from a different network. There is growing interest in integrating logical rules with labeled examples for training networks, specifically for structured outputs (Manhaeve et al., 2018; Xu et al., 2018; Fischer et al., 2019; Sun et al., 2018; Ren et al., 2018a). Xu et al. (2018); Fischer et al. (2019) convert rules on output nodes of network, to (almost differentiable) loss functions during training. The primary difference of these methods from ours is that they assume that rules are correct whereas we assume them to be noisy. Accordingly, we simultaneously correct the rules and use them to improve the classifier, whereas they use the rules as-is to train the network outputs.

A well-known framework for working with soft rules is posterior regularization (Ganchev et al., 2010) which is used in Hu et al. (2016) to train deep structured output networks while harnessing logic rules. Ratner et al. (2016) works only with noisy rules treating them as black-box labeling functions and assigns a linear weight to each rule based on an agreement objective. Our learning model is more powerful that attempts to learn a non-linear network to restrict rule boundaries rather than just weight their outputs. We presented a comparison with both these approaches in the experimental section, and showed superior performance.

To the best of our knowledge, our proposed paradigm of coupled rule-exemplar supervision is novel, and our proposed training algorithm is able to harness them in ways not possible by existing frameworks for learning from rules or noisy supervision.

## 5 CONCLUSION

We proposed a new rule-exemplar model for collecting human supervision to combine the scalability of top-level rules with the quality of instance-level labels. We show that such supervision is natural since humans typically inspect examples to code rules. Furthermore, such coupled examples provide supervision on correct firing of rules which help to denoise rules. We propose to train the classifier while jointly denoising rules via latent coverage variables imposing a soft-implication constraint on the true label. Empirically on five datasets we show that our training algorithm that performs rule-specific denoising is better than generic noise-tolerant learning. In future we plan to deploy this framework on other applications where human supervision is a scarce resource.

**Reproducibility** Code and Data for the experiments available at
https://github.com/awasthiabhijeet/Learning-From-Rules

**Acknowledgements** We thank the anonymous reviewers for their constructive feedback on this work. This research was partly sponsored by a Google India AI/ML Research Award and partly by the IBM AI Horizon Networks - IIT Bombay initiative. Abhijeet is supported by Google PhD Fellowship in Machine Learning.

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

# Supplementary Material: Learning from Rules Generalizing Labeled Exemplars

## A  POSTERIOR REGULARIZATION METHOD

We model a joint distribution $Q(y, r_1, \ldots, r_n | \mathbf{x})$ to capture the interaction among the label random variable $y$ and coverage random variables $r_1, \ldots, r_n$ of any instance $\mathbf{x}$. We use $\mathbf{r}$ to compactly represent $r_1, \ldots, r_n$. Strictly speaking, when a rule $R_j$ does not cover $\mathbf{x}$, the $r_j$ is not a random variable and its value is pinned to 0 but we use this fixed-tuple notation for clarity. The random variables $r_j$ and $y$ impose a constraint on the joint distribution $Q$: for a $\mathbf{x} \in H_j$ when $r_j = 1$, the label $y$ cannot be anything other than $\ell_j$.

$$r_j = 1 \implies y = \ell_j \quad \forall \mathbf{x} \in H_j \tag{9}$$

We can convert this into a soft constraint on the marginals of the distribution $Q$ by stating the probability of $\sum_{y \neq \ell_j} Q(y, r_j = 1 | \mathbf{x})$ should be small.

$$\min_Q \sum_j \sum_{\mathbf{x} \in H_j} \sum_{y \neq \ell_j} Q(y, r_j = 1 | \mathbf{x}) \tag{10}$$

The singleton marginals of $Q$ along the $y$ and $r_j$ variables are tied to the $P_\theta$ and $P_{j\phi}(r_j | \mathbf{x})$ we seek to learn. A network with parameters $\theta$ models the classifier $P_\theta(y | \mathbf{x})$, and a separate network with $\phi$ variables (shared across all rules) learns the $P_{j\phi}(r_j | \mathbf{x})$ distribution. The marginals of joint $Q$ should match these trained marginals and we use a KL term for that:

$$\min_{Q, \theta, \phi} \sum_{\mathbf{x} \in U \cup L} \left( KL(Q(y | \mathbf{x}); P_\theta(y | \mathbf{x})) + \sum_{j : \mathbf{x} \in H_j} KL(Q(r_j | \mathbf{x}); P_{j\phi}(r_j | \mathbf{x})) \right) \tag{11}$$

We call the combined KL term succinctly as $KL(Q, P_\theta) + KL(Q, P_\phi)$.

Further the $P_\theta$ and $P_{j\phi}$ distributions should maximize the log-likelihood on their respective labeled data as provided in Equation 1 and Equation 2 respectively.

Putting all the above objectives together with hyper-parameters $\alpha > 0, \ \lambda > 0$ we get our final objective as:

$$\min_{Q, \theta, \phi} -\alpha(LL(\theta) + LL(\phi)) + KL(Q, P_\theta) + KL(Q, P_\phi) + \lambda \sum_j \sum_{\mathbf{x} \in H_j} \sum_{y \neq \ell_j} Q(y, r_j = 1 | \mathbf{x}) \tag{12}$$

We show in Section A.1 that this gives rise to the solution for $Q$ in terms of $P_\theta$, $P_{j\phi}$ and alternately for $P_\theta$, $P_{j\phi}$ in terms of $Q$ as follows.

$$Q(y, \mathbf{r} | \mathbf{x}) \propto P_\theta(y | \mathbf{x}) \prod_{j : \mathbf{x} \in H_j} P_{j\phi}(r_j | \mathbf{x}) e^{-\lambda \delta(y \neq \ell_j \wedge r_j = 1)} \tag{13}$$

where $\delta(y \neq \ell_j \wedge r_j = 1)$ is an indicator function that is 1 when the constraint inside holds, else it is 0. Computing marginals of the above using straight-forward message passing techniques we get:

$$Q(y | \mathbf{x}) \propto P_\theta(y | \mathbf{x}) \prod_{j : \mathbf{x} \in H_j} (P_{j\phi}(1 | \mathbf{x}) e^{-\lambda \delta(y \neq \ell_j)} + P_{j\phi}(0 | \mathbf{x})) \tag{14}$$

$$Q(r_k = 1 | \mathbf{x}) \propto P_{k\phi}(1 | \mathbf{x}) \sum_y e^{-\lambda \delta(y \neq \ell_k)} P_\theta(y | \mathbf{x}) \prod_{j \neq k, \mathbf{x} \in H_j} (P_{j\phi}(1 | \mathbf{x}) e^{-\lambda \delta(y \neq \ell_j)} + P_{j\phi}(0 | \mathbf{x})) \tag{15}$$

Thereafter, we solve for $\theta$ and $\phi$ in terms of a given $Q$ as

$$\min_{\theta, \phi} -LL(\theta) - LL(\phi) - \gamma \sum_{\mathbf{x}_i \in U} \sum_{y \in \mathcal{Y}} Q(y | \mathbf{x}_i) \log P_\theta(y | \mathbf{x}_i) + \sum_{j : \mathbf{x}_i \in H_j} \sum_{r_j \in \{0,1\}} Q(r_j | \mathbf{x}_i) \log P_{j\phi}(r_j | \mathbf{x}_i) \tag{16}$$

Here, $\gamma = \frac{1}{\alpha}$. This gives rise to an alternating optimization algorithm as in the posterior regularization framework of Ganchev et al. (2010). We initialize $\theta$ and $\phi$ randomly. Then in a loop, we perform the following two steps alternatively much like the EM algorithm (Dempster et al., 1977).

**Q Computation step:** Here we compute marginals $Q(y|\mathbf{x})$ and $Q(r_j|\mathbf{x})$ from current $P_\theta$ and $P_{j\phi}$ using Equations 14 and 15 respectively for each $\mathbf{x}$ in a batch. This computation is straight-forward and does not require any neural optimization. We can interpret the $Q(y|\mathbf{x})$ as a small correction of the $P_\theta(y|\mathbf{x})$ so as to align better with the constraints imposed by the rules in Equation 3. Likewise $Q(r_j|\mathbf{x})$ is an improvement of current $P_{j\phi}$s in the constraint preserving direction. For example, the expected $r_j$ values might be reduced for an instance if its probability of $y$ being $\ell_j$ is small.

**Parameter update step:** We next reoptimize the $\theta$ and $\phi$ parameters to match the corrected $Q$ distribution as shown in Equation 16. This is solved using standard stochastic gradient techniques. The $Q$ terms can just be viewed as weights at this stage which multiply the loss or label likelihood. A pseudocode of our overall training algorithm is described in Algorithm 1.

---

**Algorithm 1** Our Joint Training Algorithm using Posterior Regularization

---

**Input:** $L, U$
   Initialize parameters $\theta, \phi$ randomly
   **for** a random training batch from $U \cup L$ **do**
      Obtain $P_\theta(y|\mathbf{x})$ from the classification network.
      Obtain $P_{j\phi}(r_j|\mathbf{x})_{j\in[n]}$ from the rule-weight network.
      Calculate $Q(y|\mathbf{x})$ using Eqn 14 and $Q(r_j|\mathbf{x})_{j\in[n]}$ using Eqn 15.
      Update $\theta$ and $\phi$ by taking a step in the direction to minimize the loss in Eqn 16.
   **end for**
**Output:** $\theta$ , $\phi$

---

## A.1 Proof: Alternating solution for Optimization Objective in Eqn 12

Treat each $Q(y, \mathbf{r})$ as an optimization variable with the constraint that $\sum_{y,\mathbf{r}} Q(y, \mathbf{r}) = 1$. We express this constraint with a Langrangian multiplier $\eta$ in the objective. Also, define a distribution

$$P_{\theta,\phi}(y,\mathbf{r}|\mathbf{x}) = P_\theta(y|\mathbf{x}) \prod_{j:\mathbf{x}\in H_j} P_{j\phi}(r_j|\mathbf{x})$$

It is easy to verify that the KL terms in our objective 12 can be collapsed as $KL(Q; P_{\theta,\phi})$. The rewritten objective (call it $F(Q, \theta, \phi)$ ) is now:

$$-\alpha(LL(\theta) + LL(\phi)) + \sum_{\mathbf{x}} KL(Q(y,\mathbf{r}|\mathbf{x}), P_{\theta,\phi}(y,\mathbf{r}|\mathbf{x}))$$
$$+\lambda \sum_j \sum_{\mathbf{x}\in H_j} \sum_{y\neq\ell_j} Q(y, r_j = 1|\mathbf{x}) + \eta(1 - \sum_{y,\mathbf{r}} Q(v,r)) \tag{17}$$

Next we solve for $\frac{\partial F}{\partial Q(y,\mathbf{r})} = 0$ after expressing the marginals in their expanded forms: e.g. $Q(y, r_j|\mathbf{x}) = \sum_{r_1,\dots,r_{j-1},r_{j+1},\dots,r_n} Q(y, r_1, \dots, r_n|\mathbf{x})$. This gives us

$$\frac{\partial F}{\partial Q(y,\mathbf{r})} = \quad \log Q(y, \mathbf{r}) - \log P_{\theta,\phi}(y, \mathbf{r}|\mathbf{x})$$
$$+ \sum_{j:\mathbf{x}\in H_j} \lambda\delta(y \neq \ell_j, r_j = 1) + \eta + 1$$

Equating it to zero and substituting for $P_{\theta,\phi}$ we get the solution for $Q(y, \mathbf{r})$ in Equation 13.

The proof for the optimal $P_\theta$ and $P_{j\phi}$ while keeping $Q$ fixed in Equation 17 is easy and we skip here.

# B  LIST OF RULES

We provide a list of rules for each task type.

| Rule | Example | Class |
|---|---|---|
| `( |^)(where)[^\w]* (\w+ ){0,1}` `(was|is)[^\w]*( |\$)` | Where is Trinidad ? | Location |
| `( |^)(which|what)[^\w]* (\w+ ){0,1}` `(play|game|movie|book)[^\w]*( |$)` | What book is the follow-up to Future Shock ? | Entity |
| `( |^)(what)[^\w]* (\w+ ){0,1}` `(part|division|ratio|percentage)` `[^\w]*( |$)` | Of children between the ages of two and eleven , what percentage watch " The Simpsons " ? | Numeric |
| `( |^)(who|who)[^\w]* (\w+ ){0,1}` `(found|discovered|made|built` `|build|invented)[^\w]*( |$)` | Who invented volleyball ? | Human |

Table 4: Sample rules for TREC Question Classification. Rule fires if the regex matches

| Rule | Example | Class |
|---|---|---|
| `( |^)(free)[^\w]*` `([^\s]+ )*(price)[^\w]*` `([^\s]+ )*(call)[^\w]*( |$)` | Free video camera phones with Half Price line rental for 12 mths and 500 cross ntwk mins 100 txts. Call MobileUpd8 08001950382 or Call2OptOut/674 | Spam |
| `( |^)(guranteed)[^\w]* ([^\s]+ )*` `(gift\.|gift)[^\w]*( |$)` | Great News!  Call FREEFONE 08006344447 to claim your guaranteed å£1000 CASH or å£2000 gift. | Spam |
| `( |^)(can't)[^\w]*` `(\w+ ){0,1}(talk)[^\w]*( |$)` | sry can't talk on phone, with parents | NotSpam |
| `( |^)(that's)[^\w]*` `(\w+ ){0,1}(fine!|fine)[^\w]*( |$)` | Yeah, that's fine! It's å£6 to get in, is that ok? | NotSpam |

Table 5: Sample rules for Spam Classification. Rule fires if the regex matches

| Rules | Class |
|---|---|
| capital-gain $> 6849$ | $> 50K$ |
| education-num $> 12$ AND marital-status = Never-married AND native-country = United-States AND occupation = Exec-managerial | $> 50K$ |
| marital-status = Separated AND hours-per-week $\leq 41$ | $\leq 50K$ |
| education-num $\leq 12$ AND native-country = United-States AND age $\leq 30$ | $\leq 50K$ |

Table 6: Sample rules for census dataset. Rule fires if all clauses are True

| Rule | Example | Class |
|------|---------|-------|
| `( \|^)[^\w]*`
`(within\|near\|next\|close\|nearby\|`
`around\|around)[^\w]*([^\s]+ ){0,2}`
`(here\|city\|miles\|mile)`
`*[^\w]*( \|$)` | any kid friendly restaurants around here | Location |
| WordLists:

`cuisine1a=['italian','american',`
`'japanese','spanish','mexican',`
`'chinese','vietnamese','vegan']`

`cuisine1b=['bistro','delis']`

`cuisine2=['barbecue','halal',`
`'vegetarian','bakery']` | can you find me some chinese food | Cuisine |
| `([0-9]+\|few\|under [0-9]+) dollar` | i need a family restaurant with meals under 10 dollars and kids eat | Price |
| `((high\|highly\|good\|best\|top\|`
`well\|highest\|zagat)`
`(rate\|rating\|rated))\|`
`((rated\|rate\|rating)`
`[0-9]* star)\|([0-9]+ star)` | where can i get the highest rated burger within ten miles | Rating |
| `((open\|opened) (now\|late))\|`
`(still (open\|opened\|closed\|close))`
`\|(((open\|close\|opened\|closed)`
`\w+([\s]\| \w* \| \w* \w* ))*[0-9]+`
`(am\|pm\|((a\|p) m)\|hours\|hour))` | where is the nearest italian restaurant that is still open | Hours |
| `(outdoor\|indoor\|group\|romantic\|`
`family\|outside\|inside\|fine\|`
`waterfront\|outside\|private\|`
`business\|formal\|casual\|rooftop\|`
`(special occasion))`
`([\s]\| \w+ \| \w+ \w+ )dining` | i want to go to a restaurant within 20 miles that got a high rating and is considered fine dining | Amenity |
| `[\w+ ]{0,2}(palace\|cafe\|bar\|`
`kitchen\|outback\|dominoes)` | is passims kitchen open at 2 am | Restaurant Name |
| `wine\|sandwich\|pasta\|burger\|`
`peroggis\|burrito\|`
`(chicken tikka masala)\|`
`appetizer\|pizza\|wine\|`
`cupcake\|(onion ring)\|tapas` | please find me a pub that serves burgers | Dish |

Table 7: Sample rules for MIT-R dataset. Rule fires if the regex matches or sentence contains a word found in the provided word lists.

## C  HYPERPARAMETERS

Across all experiments we use Adam optimizer with default values of $\beta_1, \beta_2$, and $\epsilon$. Dropout of 0.8 (keep probability) was used in the feed forward layers. All the models were trained for a maximum of 100 epochs and early stopping was used based on a validation set. Best model on the validation set was evaluated on the test set. Each experiment was run with 10 random initializations. A list of hyperparameters used in our experiments is provided below.

| | Noise-tolerant | Snorkel-Noise-Tolerant | Post. Reg. | implication | L+Usnorkel | L+Umaj |
|---|---|---|---|---|---|---|
| | | | Question Classification | | | |
| $\gamma$ | 0.001 | 0.1 | 0.001 | 0.1 | 0.01 | 0.001 |
| $q$ | 0.9 | 0.6 | - | - | - | - |
| $lr$ | | | 0.0003 | | | |
| $bs$ | | | 32 (16 for Only-L) | | | |
| | | | MIT-R | | | |
| $\gamma$ | 0.01 | 0.001 | 0.01 | 0.1 | 0.05 | 0.01 |
| $q$ | 0.6 | 0.6 | - | - | - | - |
| $lr$ | | | 0.0003 | | | |
| $bs$ | | | 64 (32 for Only-L) | | | |
| | | | YouTube | | | |
| $\gamma$ | 0.003 | 0.5 | 0.1 | 0.2 | 0.5 | 0.003 |
| $q$ | 0.6 | 0.6 | - | - | - | - |
| $lr$ | | | 0.0003 | | | |
| $bs$ | | | 32 (16 for Only-L) | | | |
| | | | SMS | | | |
| $\gamma$ | 0.1 | 0.1 | 0.001 | 0.3 | 0.5 | 0.1 |
| $q$ | 0.6 | 0.6 | - | - | - | 0.1 |
| $lr$ | | | 0.0001 | | | |
| $bs$ | | | 32 (16 for Only-L) | | | |
| | | | Census | | | |
| $\gamma$ | 0.5 | 0.1 | 0.001 | 0.1 | 0.01 | 0.5 |
| $q$ | 0.1 | 0.6 | - | - | - | 0.5 |
| $lr$ | | 0.0001 | | | 0.0003 | |
| $bs$ | | | 64 (16 for Only-L) | | | |

Table 8: Hyperparameters for various methods and datasets. $bs$ refers to the batch size and $lr$ refers to the learning rate. For Only-L baseline smaller batch size was used considering the smaller size of $L$ set.

| | Question | MIT-R | YouTube | SMS | Census |
|---|---|---|---|---|---|
| $meta\_lr$ | 0.01 | 0.0001 | 0.001 | 0.0001 | 0.0001 |
| $lr$ | | 0.0003 | | 0.0001 | 0.0003 |
| $bs$ | 32 | 64 | 32 | 32 | 64 |

Table 9: Meta-learning rate, learning rate and batch size used for L2R (Ren et al., 2018b) for various datasets

