# OpenReview forum: "Learning from Rules Generalizing Labeled Exemplars"
_ICLR.cc/2020/Conference — Accept (Spotlight)_

### Official Review · AnonReviewer1 · 2019-10-21
**Official Blind Review #1**

**Rating:** 6

**Review:**

In case of a lack of labeled data, human-designed rules can be used to label the unlabelled data. This paper proposes a better rule-based labeling method by restricting the coverage of the rule, which is based on the assumption that the rules can be applied to a local region but can not be 'over-generalized' to the whole sample space. The coverage of the rule is represented by a conditional distribution, which is parameterized as a neural network and jointly learned with the classifier network.

I think this paper is tackling an important problem in machine learning, and the proposed idea is novel and interesting. I vote for weak acceptance because there are still some technical points that are not well-addressed enough:

First, although the intuition of this model makes a lot of sense to me, the construction of the loss function is quite heuristic, with a lot of terms simply summing together, making it hard to judge which components are most important for the final results. A more principled and integrated framework like EM could be more convincing to me.

Second, it seems the unlabelled data is only used in the causal constraint term (the last term in Eqn 5) and it is controlled by a coefficient \gamma. It is a bit unclear to me whether the unlabelled data is fully utilized while it only constraints the causal relation, as one can also use labeled data for constraining the causal relation. Also, why not include labeled data for this constraint regularization?

Another minor question is after the two networks are trained, will you only use the learned classifier for test data, or, do you also use the conditional distribution in the testing phase and compute an expectation of the predicted class? and why?

Also, what's the purpose of section 6 in the appendix?

I general I think the idea of learning a conditional distribution to constrain the use of rules is an interesting and novel idea. The paper can be further improved if the algorithm can be more principled.


---- after reading the response ---

Thanks for answering the questions. I believe some of these explanations can be added to the final version to improve clarity. My score does not change, but overall I advocate to accept this paper.


**Experience Assessment:**

I have published one or two papers in this area.

**Review Assessment: Checking Correctness Of Derivations And Theory:**

I did not assess the derivations or theory.

**Review Assessment: Checking Correctness Of Experiments:**

I assessed the sensibility of the experiments.

**Review Assessment: Thoroughness In Paper Reading:**

I read the paper at least twice and used my best judgement in assessing the paper.

---

> ### Author Response · Authors · 2019-11-13
> **Response to Reviewer #1**
>
> Thank you for providing valuable feedback on our work. We have addressed your comments/questions below:
>
> > First, although the intuition of this model makes a lot of sense to me, the construction of the loss function is quite heuristic, with a lot of terms simply summing together, making it hard to judge which components are most important for the final results. A more principled and integrated framework like EM could be more convincing to me.
>
> Our first attempt was indeed a principled EM framework which we called the Posterior regularization (PR) method that we derive in detail in the Supplementary.  Unfortunately, we found that this EM formulation performed worse than the only-L baseline in two of the five datasets (Table 2). That is what led us to our current non-EM formulation which provided higher accuracy while being (frustratingly :) simpler.  Also, it was more robust to hyper-parameter selection and initializations than the EM formulation. Finally, if you compare our ImplyLoss objective (Eq 5) with  EM’s objective (Eq 14), the two labeled loss terms are identical.  The only difference is in the term involving unlabeled data.  In EM,  the KL term biases the model to match the estimated distribution by the teacher which is changing and could be incorrect.  In contrast, the ImplyLoss does not introduce any such bias as we discuss in Section 2.1 (below Table 1).
>
> > Second, it seems the unlabelled data is only used in the causal constraint term (the last term in Eqn 5) and it is controlled by a coefficient \gamma. It is a bit unclear to me whether the unlabelled data is fully utilized while it only constraints the causal relation, as one can also use labeled data for constraining the causal relation.
>
> Yes, unlabelled data is only used in the last term of Eqn 5. This term allows full utilization of the correctly generalized unlabelled data as follows: When P(r_j =1|x_i) is close to one, the causal term reduces to log-likelihood of labeling x_i as l_j, allowing us to treat x_i as a labeled instance. The same causal term allows us to ignore wrongly generalized instances as follows:  when P(r_j =1|x_i) is close to zero, the gradient on the label parameters (\theta) is vanishingly small.
>
> > Also, why not include labeled data for this constraint regularization?
>
> Empirically, we found no difference between including labeled data in the causal term or not.
> The causal term when fitted with the clean labels for ‘y_i’ and r_{ji} (when available) reduce to the likelihood term LL(\theta). We did not want to double count labeled instances and distort the training distribution.
>
> > Another minor question is after the two networks are trained, will you only use the learned classifier for test data, or, do you also use the conditional distribution in the testing phase and compute an expectation of the predicted class? and why?
>
> We only use the classification network (parametrized by \theta) for test data. The conditional distribution is only applicable when a rule covers the instance. However, an example in the test set may not be covered by any of the rules (For one of our datasets coverage of rules is only 14% ).  We also tried to predict the 'y' that minimizes a joint score over y and r_ji conditionals. But in practice we did not see much difference.
>
> > Also, what's the purpose of section 6 in the appendix?
>
> Section 6 describes our attempt of an EM-based framework (Algorithm 1, Pg. 13) for this task. We compare implication loss with this alternative approach of jointly learning the classifier and rule network by imposing the same causal constraints in a different way.

---

### Official Review · AnonReviewer2 · 2019-10-22
**Official Blind Review #2**

**Rating:** 8

**Review:**

This paper proposes a novel semi-supervised learning paradigm where the algorithm learns from both clean instance-level labels and noisy rule-level labels, and also a simple but effective algorithm as solution. The proposed algorithm employs a set of latent coverage variables to bridge two kinds of supervisions and uses a soft causal constraint on the coverage variables to denoise the noisy labels. Empirically the paper demonstrates the effectiveness of the proposed algorithm with consistent improvements over several baselines on a wide range of classification tasks.

The idea of using macro-level noisy labels as part of the supervision is novel, and it could potentially trigger a paradigm shift on many research areas in machine learning. The proposed methodology is clean but effective, with extensive experimental support. Therefore I vote for accepting this submission.

Minor problems

(1) Abuse of notation \phi in section 2.
(2) "... from traing the classifier ..." in page 4.


More (further) questions

(1) Since each rule can be regarded as experts or weak learners, how is this work related to learning strong learners from weak learners (boosting/ensemble)?
(2) Is it possible that the algorithm can incorporate more information of the rules, for example, the structure of the logical formulas?
(3) Is it possible to generalize the idea to RL?

**Experience Assessment:**

I have read many papers in this area.

**Review Assessment: Checking Correctness Of Derivations And Theory:**

I carefully checked the derivations and theory.

**Review Assessment: Checking Correctness Of Experiments:**

I assessed the sensibility of the experiments.

**Review Assessment: Thoroughness In Paper Reading:**

I read the paper thoroughly.

---

> ### Author Response · Authors · 2019-11-13
> **Response to Reviewer #2**
>
> Thank you for providing valuable feedback on our work. We have addressed your comments/questions below:
>
> > Minor Problems
>
> We have fixed the notations and other typing errors in the revised version.
>
> > Since each rule can be regarded as experts or weak learners, how is this work related to learning strong learners from weak learners (boosting/ensemble)?
>
> One big difference is that most rules cover only a small number of examples, and for a given example only a small number of rules cover them (Table 1). This is unlike the setting of typical “weak to strong learners” framework where all weak learners predict labels for all examples.
>
> > Is it possible that the algorithm can incorporate more information of the rules, for example, the structure of the logical formulas?
>
> In this work, we wanted to treat rules as black-box functions. Exploiting the logical formulas should help but will entail more application-specific encodings.
>
> > Is it possible to generalize the idea to RL?
>
> Yes, our method can be extended to RL in the imitation learning setting in order to reduce the amount of human-generated data used in imitation learning. A rule in this scenario will be a partial policy, i.e., it will map some states to an action but may not cover other states. During training, the state space could either be sampled completely randomly or by following the policy formed by applying the current rule weights to each rule, or a combination.

---

> > ### Comment · AnonReviewer2 · 2019-11-15
> > **Thanks for the response**
> >
> > Thanks for answering the questions.

---

### Official Review · AnonReviewer4 · 2019-11-01
**Official Blind Review #4**

**Rating:** 6

**Review:**

The paper addresses the problem that labelled data is often unavailable in the quantities required to train effective models. It deals with classification problems, and proposes a method to obtain more (but weaker) labels data with minimal involvement from human labellers, by asking them to generalize their labelling decisions into rules and then learning restrictions on those rules to avoid learning incorrectly generalized labels. The motivating observation is that human labellers are often able to make such generalizations in much less time than it would take them to apply that rule to a large dataset themselves. This is an interesting idea, especially for cases where labelling capacity is limited. The point being made about the labelling noise not being random in this situation is an interesting one - it might be worth exploring this notion further on its own also in contexts where the source of the noise is unknown.

The presentation of the implementation the authors choose for their proposed approach is clear, and the implementation is sensible. The experimental section includes comparisons to a number of alternative methods, and the authors find that their method outperforms all others, including recent methods for combining (noisy) rule-based labels and (clean) human-sourced labels.

I would argue for accepting this paper. It studies an interesting question, which if answered has the potential to make access to machine learning solutions to certain types of problem significantly cheaper and therefore more widespread. The experiments are well-chosen and show that, depending on the data available and the task, significant gains can be made using the proposed method.

Some remarks on how the paper could become stronger: The type of problem that can be assessed with the proposed method seems to be fairly specific: most tasks studied are classification of natural language utterances. That is a natural class of tasks, since it is easy to imagine how labellers can formulate rules. However, it would have been very interesting if the authors had found ways to allow for more diversity here.  In general, I have the impression that there are more interesting ideas and results to be found in the direction explored by this paper - what about, e.g., allowing the classifier to add rules of its own?

The paper would benefit from some general editing with regards to appearance; for example, the supplementary material sections continue the regular section numbering, instead of having their own; the images are missing captions, and sometimes have somewhat unorthodox axis tick labelling.

**Experience Assessment:**

I have read many papers in this area.

**Review Assessment: Checking Correctness Of Derivations And Theory:**

I assessed the sensibility of the derivations and theory.

**Review Assessment: Checking Correctness Of Experiments:**

I assessed the sensibility of the experiments.

**Review Assessment: Thoroughness In Paper Reading:**

I read the paper at least twice and used my best judgement in assessing the paper.

---

> ### Author Response · Authors · 2019-11-13
> **Response to Reviewer #4**
>
> Thank you for providing valuable feedback on our work. We have addressed your comments/questions below:
>
> > Some remarks on how the paper could become stronger: The type of problem that can be assessed with the proposed method seems to be fairly specific: most tasks studied are classification of natural language utterances. That is a natural class of tasks, since it is easy to imagine how labellers can formulate rules. However, it would have been very interesting if the authors had found ways to allow for more diversity here.  In general, I have the impression that there are more interesting ideas and results to be found in the direction explored by this paper - what about, e.g., allowing the classifier to add rules of its own?
>
> Extending this work to more diverse tasks is our goal in future research effort. Allowing the classifier to add rules of its own seems an interesting direction.
>
> > The paper would benefit from some general editing with regards to appearance; for example, the supplementary material sections continue the regular section numbering, instead of having their own; the images are missing captions, and sometimes have somewhat unorthodox axis tick labelling.
>
> We have fixed these in the revision.

---

### Author Response · Authors · 2019-10-18
**Link to anonymized code**

https://github.com/iclrLFRGLE/iclrLFRGLE

---

### Author Response · Authors · 2019-11-13
**General Response**

We thank the reviewers for constructive and insightful feedback on our work. We have addressed each reviewer’s comments/questions individually.

Based on the reviewers’ feedback, we have made the following changes in the revised version:
1. Added missing captions to figures and tables
2. Modified section numbering in supplementary
3. Fixed notational and typing errors
4. Combined the tables of hyperparameters in supplementary for better appearance

---

### Decision · Program_Chairs · 2019-12-19

**Decision:**

Accept (Spotlight)

**Comment:**

The paper addresses the problem of costly human supervision for training supervised learning methods.
The authors propose a joint approach for more effectively collecting supervision data from humans, by extracting rules and their exemplars, and a model for training on this data.
They demonstrate the effectiveness of their approach on multiple datasets by comparing to a range of baselines.

Based on the reviews and my own reading I recommend to accept this paper.
The approach makes intuitively a lot of sense and is well explained.
The experimental results are convincing.